# Case Study: Development of the CNN Model Considering Teleconnection for Spatial Downscaling of Precipitation in a Climate Change Scenario

Jongsung Kim [1], Myungjin Lee [1,*], Heechan Han [2], Donghyun Kim [3], Yunghye Bae [3] and Hung Soo Kim [3]

1 Institute of Water Resources System, Inha University, Incheon 22201, Korea; kjjs0308@naver.com
2 Blackland Research and Extension Center, Texas A&M AgriLife, Temple, TX 76502, USA; heechan.han@ag.tamu.edu
3 Department of Civil Engineering, Inha University, Incheon 22201, Korea; yesdktpdi@naver.com (D.K.); yhbaebae@gmail.com (Y.B.); sookim@inha.ac.kr (H.S.K.)
* Correspondence: lmj3544@naver.com

**Abstract:** Global climate models (GCMs) are used to analyze future climate change. However, the observed data of a specified region may differ significantly from the model since the GCM data are simulated on a global scale. To solve this problem, previous studies have used downscaling methods such as quantile mapping (QM) to correct bias in GCM precipitation. However, this method cannot be considered when certain variables affect the observation data. Therefore, the aim of this study is to propose a novel method that uses a convolution neural network (CNN) considering teleconnection. This new method considers how the global climate phenomena affect the precipitation data of a target area. In addition, various meteorological variables related to precipitation were used as explanatory variables for the CNN model. In this study, QM and the CNN models were applied to calibrate the spatial bias of GCM data for three precipitation stations in Korea (Incheon, Seoul, and Suwon), and the results were compared. According to the results, the QM method effectively corrected the range of precipitation, but the pattern of precipitation was the same at the three stations. Meanwhile, for the CNN model, the range and pattern of precipitation were corrected better than the QM method. The quantitative evaluation selected the optimal downscaling model, and the CNN model had the best performance (correlation coefficient (CC): 69% on average, root mean squared error (RMSE): 117 mm on average). Therefore, the new method suggested in this study is expected to have high utility in forecasting climate change. Finally, as a result of forecasting for future precipitation in 2100 via the CNN model, the average annual rainfall increased by 17% on average compared to the reference data.

**Keywords:** climate change; convolution neural network; spatial downscaling; teleconnection; quantile mapping

## 1. Introduction

The magnitude and frequency of extreme natural disasters such as floods and droughts are increasing worldwide as climate change influences weather patterns [1–4]. The Intergovernmental Panel on Climate Change (IPCC) projected climate variables such as precipitation and temperature for the next 100 years to warn about the impact of climate change on flooding and droughts in many regions around the globe [5–9]. Furthermore, climate variability has impacted water resource management as well as environmental, socioeconomic, and agricultural ecosystems [10–12].

The IPCC has provided regular scientific assessments representing knowledge on climate change, its causes, and potential future risk. The representative assessment reports (AR) were published in 2007 (AR4) and 2014 (AR5), and a new AR6 will be released in 2022. Based on the climate change scenario data provided by these AR, global climate models (GCMs) and global atmosphere–ocean circulation models simulate future climate variability.

The models consider complex interactions between the atmosphere–ocean–land surfaces affected by the greenhouse effect, solar energy, volcanic eruptions, and other natural phenomena [11]. In addition, the GCMs provide projected meteorological variables, including precipitation and temperature, for the next 100 years worldwide. Generally, hydrological models are used to simulate the hydrological impact of climate change using the GCM datasets as inputs [13–15]. However, the GCM outputs cannot be used directly as input to the hydrological models because of their inconsistent and low spatial resolution [15]. Therefore, spatial downscaling must be conducted to use a GCM at a regional scale for proper hydrological simulations [11].

There are two primary methods for downscaling GCM outputs: dynamic downscaling and statistical downscaling methods. The active downscaling methods have physical meaning, but they also have limitations in terms of expensive computational processes. Dynamic downscaling aims to translate the large-scale weather features from GCM outputs into higher-resolution data using regional climate models [16]. Yang et al. [3] presented an improved dynamical downscaling method with GCMs bias correction, and they found that this correction significantly reduced the uncertainties of downscaled meteorological variables such as temperature, wind, moisture, and precipitation. Hermans et al. [17] found that dynamical downscaling can dramatically impact meteorological forces and sea-level ocean changes when using high-resolution regional models forced with output from GCMs.

In contrast, statistical downscaling methods have fewer computational processes [15]. Quantile mapping is a representative statistical downscaling method that has been used in many studies to correct for biases in GCM data [18–21]. The quantile mapping method corrects biases in climate models for observational data based on the cumulative distribution functions (CDFs). Xu and Wang [18] analyzed spatial downscaling for extreme temperatures in China using various GCM data from CMIP5, and quantile mapping was used as a spatial downscaling method for bias correction of GCM data. As a result, the bias for most regions was decreased, and it showed that the data corrected through quantile mapping were significantly improved compared to the original GCM data. In addition, the statistical relationship between regional-scale meteorological predictors and circulation features typically showed a form of the regression model [13]. For example, Landman et al. [22] used a perfect prognosis approach to downscale the GCM outputs from coarse resolution to catchment level to provide a categorized streamflow forecast, and they found that the proposed method provided successful predictions of streamflow categories. Tisseuil et al. [23] used four statistical downscaling methods—namely generalized linear and additive models, aggregated boosted trees, and multi-layer perceptron neural networks—to estimate summary flow statistics enhanced by characterized precipitation and evaporation. Downscaling methods have been widely used in various research fields such as hydrology and meteorology [24–32]. However, GCMs contain significant systematic errors that significantly affect model performance and sometimes cause considerable uncertainty in modeling outcomes. The downscaling process also reflects these uncertainties, which involves the quality of downscaling results. Therefore, the bias of the downscaling method is a significant cause of uncertainty in climate change scenarios and hydrological modeling performance [33].

As computing hardware systems and algorithms improve, data-driven methods such as machine learning and deep learning models have been widely used to downscale GCM products. The data-driven models have shown high accuracy in terms of the downscaling results of various meteorological variables related to precipitation and temperature [34–38]. In addition, many types of data-driven models have been applied to downscale GCM datasets, including artificial neural networks (ANNs), support vector machines (SVMs), long-short term memory model (LSTM), and the K-nearest neighbor (KNN) method. For example, Chen et al. [39] applied the smooth support vector machine and ANN model to predict daily precipitation using GCMs. Ahmadi et al. [40] used the SVM, KNN, and ANN models for precipitation prediction and demonstrated that data-driven models can achieve high performance when downscaling precipitation. Reportedly, seasonal patterns of local climate variables such as precipitation and temperature are highly correlated with

global-scale climate variability [41–43]. Previous studies have identified several locations where the climate features of the specific local area are linked, and they showed high accuracy in the prediction of climate variables (i.e., precipitation and temperature) when considering global teleconnection indices in their model.

Many previous studies that used the traditional downscaling method still have limitations since downscaled precipitation has many biases (under- or overestimations) compared to ground-based observation data [44–46]. To fill the gap associated with this limitation, this study aims to develop a deep learning-based method for downscaling the precipitation products generated from GCMs and compare it with the traditional method. In this study, we evaluate a convolutional neural network (CNN) model, which is a typical deep learning algorithm for downscaling precipitation in the Model for Interdisciplinary Research on Climate (MIROC6). In addition, to further strengthen the proposed model's performance, datasets from grids considering global teleconnections as well as local datasets were used as input data for the downscaling model.

## 2. Materials and Methods

### 2.1. Study Area

The Republic of Korea is a peninsula at 33–38° N and 124–131° E; more than half of the total area is mountainous terrain. Since Korea's rainfall characteristics are affected by its monsoon climate, rainfall is concentrated in the summer (June–October), and regional variations are extreme depending on the topographical characteristics. To continuously manage water resources considering the regional rainfall characteristics, the Korea Meteorological Administration has been monitoring meteorological phenomena (including precipitation) for more than 30 years, using 95 meteorological observation systems. The Incheon, Seoul, and Suwon stations, which are included in the metropolitan area, were selected as the target areas in this study. These stations are included in one grid of GCM data (MIROC 6). Figure 1 shows the MIROC6 grids, which cover the entire study area, and the precipitation stations considered in this study.

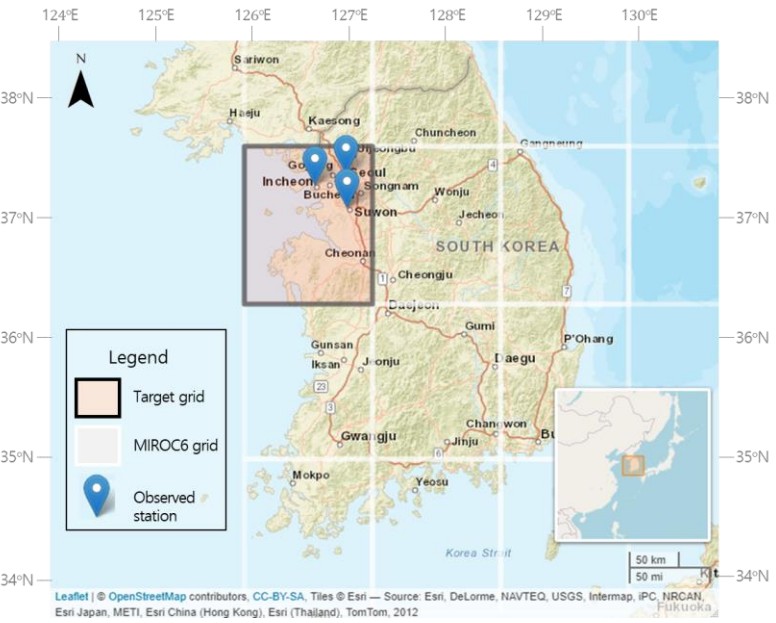

**Figure 1.** Target area and GCM data.

### 2.2. Datasets

This study used the MIROC6 climate change scenario provided by the University of Tokyo Center for Climate System Research and the National Institute for Environmental Studies. The MIROC6 data has a spatial resolution of 1.41° × 1.41° and covers the globe, with 256 × 128 cells in total. The MIROC6 data follows the IPCC 6th report and is based on

the Shared Socioeconomic Pathways (SSP) scenario. The SSP scenarios are divided into four scenarios (SSP1-2.6, SSP2-4.5, SSP3-7.0, and SSP5-8.5) according to future climate change adaptation efforts, and they consider various factors such as population, economy, welfare, ecosystem, resources, technological development, and policy. In this study, we used historical data and the SSP5-8.5 data to identify extreme scenarios. Here, the SSP5-8.5 scenario assumes that city-centered indiscreet development will expand. Table 1 summarizes the features of the GCM data used in this study.

**Table 1.** GCM data used in this study.

| Data Type | Description | Resolution (Lon. × Lat.) | Cell Size | Period (Monthly) |
|---|---|---|---|---|
| Historical data | - | 1.41° × 1.41° | 256 × 128 | 1850-01-01–2014-12-31 |
| SSP 8.5 scenario data | Extreme scenario | 1.41° × 1.41° | 256 × 128 | 2015-01-01–2100-12-31 |

To check the spatial variability according to the downscaling method, precipitation data at three stations (Incheon, Seoul, and Suwon) were used, and these stations are included in one grid of GCM data. The Incheon and Seoul stations have been observed since August 1904 and October 1907, respectively, and the Suwon station has been observed since January 1964. These data were used as reference data for the spatial downscaling of GCM data. Table 2 summarizes the features of the precipitation station data used in this study.

**Table 2.** Precipitation station data used in this study.

| Station Name | Location | Elevation | Observation Start Date | Institutions |
|---|---|---|---|---|
| Incheon | lat: 37.4777° lon: 126.6249° | 68.99 m | 1904-08-29 | Korea Meteorological Administration |
| Seoul | lat: 37.5714° lon: 126.9658° | 85.67 m | 1907-10-01 | Korea Meteorological Administration |
| Suwon | lat: 37.2723° lon: 126.9853° | 34.84 m | 1964-01-01 | Korea Meteorological Administration |

For this study, both the reference data and GCM data were collected monthly. Figure 2 shows the time series compared with the reference data and the GCM data during a historical period. In Figure 2, the dotted line represents the range of the GCM precipitation data. That data is in the range 0–743 mm. For the Incheon station, the precipitation data ranges from 0 to 1364 mm, for Seoul, it varies from 0 to 1105 mm, and for Suwon, the range is 0–967 mm. Here, the range of values is clearly different. In addition, the monthly precipitation patterns differ for each station compared with the GCM data. Table 3 summarizes the statistics for each precipitation data type; here, reference data are all monthly data.

**Table 3.** Statistics for each precipitation data type.

| Data Type | Min | Max | Mean | Standard Deviation |
|---|---|---|---|---|
| GCM data | 0 mm | 743 mm | 106.9 mm | 113.3 mm |
| Incheon station | 0 mm | 1364 mm | 111.3 mm | 147.6 mm |
| Seoul station | 0 mm | 1105 mm | 94 mm | 119.2 mm |
| Suwon station | 0 mm | 967 mm | 111 mm | 137.8 mm |

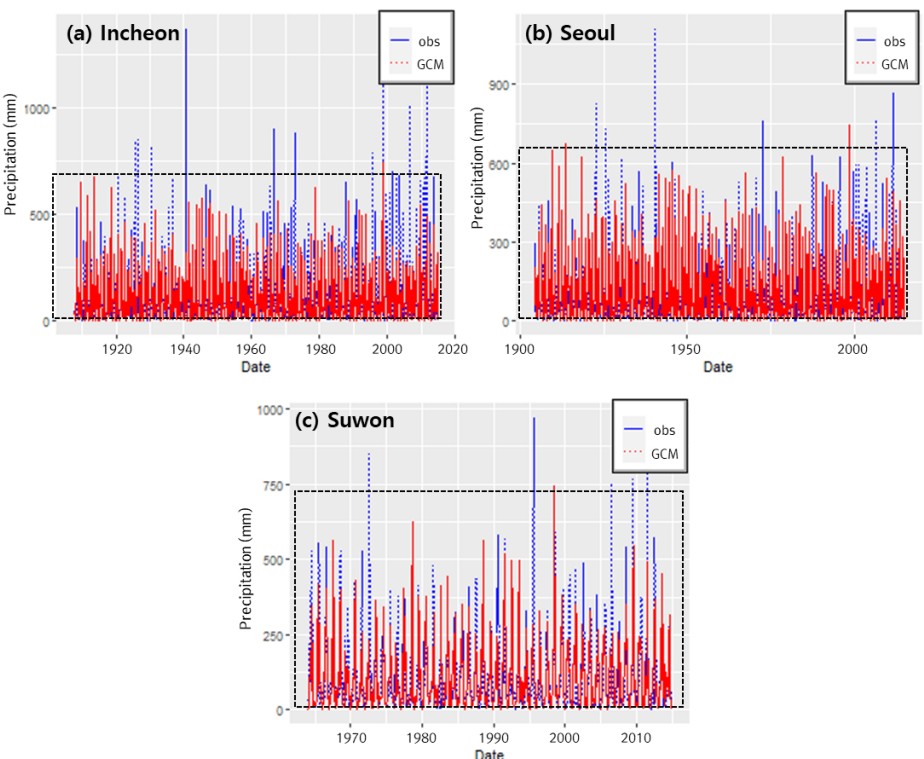

**Figure 2.** Monthly time series comparing the reference data and GCM data in the historical period. (The black dotted line: the range of the GCM precipitation data).

### 2.3. Downscaling Methods

#### 2.3.1. Quantile Mapping

Since the spatial resolution of the climate change scenario data is too large, there is a difference between the GCM precipitation and the actual precipitation at each rainfall station (see Figures 1 and 2). Quantile mapping (QM) is a representative spatial downscaling method; many previous studies have used it to calibrate GCM data [18–21]. QM has various methods, including a method that fits a theoretical distribution to both observed and modeled time series, a method that fits parametric transformations to the quantile–quantile relationship between the observed and modeled values, and a smoothing spline method for the quantile–quantile plot of the observed and modeled time series. The QM method used in this study is a theoretical distribution method that fits the probability distribution of the GCM data to that of the observed data to reduce the error between the GCM and the observed data [47]. The cumulative probability of the GCM data is converted into a value that corresponds to the cumulative probability of the observation data using the quantile mapping method. In addition, the probability distribution of the corrected GCM data has the same probability distribution as the observation data; this process is expressed as Equation (1):

$$Z_j = F_{oj}^{-1}\big(F_{sj}(\hat{Y}_j)\big) \tag{1}$$

where $\hat{Y}_j$ is the GCM data before correction, $Z_j$ is the calibrated GCM data, $F_{sj}$ is the cumulative probability distribution of the original GCM data, and $F_{oj}$ is the cumulative probability distribution of the observation data.

#### 2.3.2. The CNN Model Considering Teleconnection

This study developed a CNN model for the spatial downscaling of precipitation using various meteorological data. We conducted a teleconnection analysis to find factors that directly and indirectly affect climate information on a global scale. Figure 3 shows the procedure used by the CNN model in this study, which includes the following two phases:

(1) extraction of the highly correlated grid; and (2) CNN modeling for spatial downscaling. During phase 1, Sea Surface Temperature (SST) data were analyzed for temporal and spatial correlations using reference precipitation data. SST data have primarily been used to identify climatic phenomena between the ocean and atmosphere, such as El Niño and La Niña. During phase 2, the CNN models were trained by the explanatory variables that are the extracted SST data and the meteorological data of the corresponding grid. Here, the dataset was divided into a training set (1904-08~1989-12) and a test set (1990-01~2014-12); the training set was used to train the CNN model, and the test set was used to evaluate the model. This study was divided into three downscaling steps. The first step was the model training, which used a training dataset for the CNN model and the QM method. The second step was model evaluation and selection of an optimal model. The third step was precipitation forecasting using the SSP scenario.

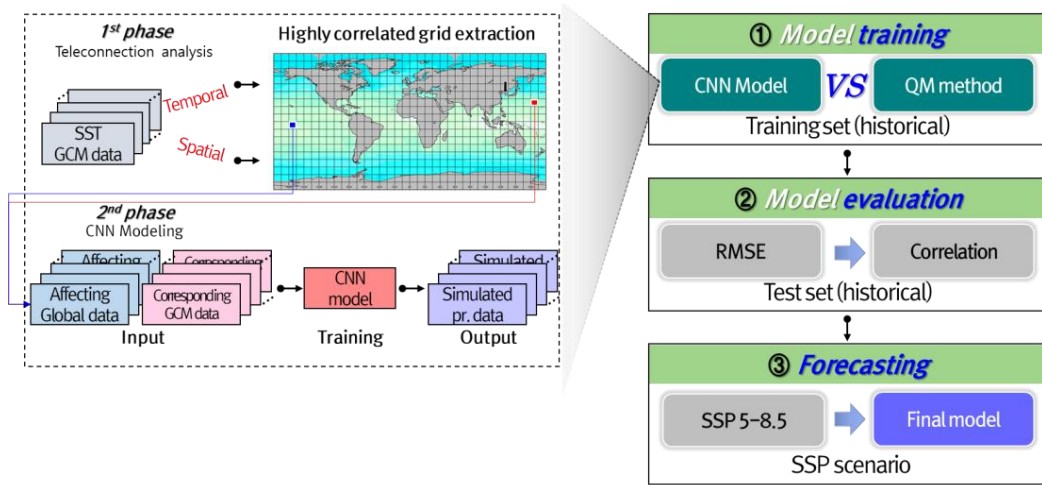

**Figure 3.** The procedure of the CNN model and a flow chart of this study.

Convolutional Neural Network

Deep learning can be defined as an upgraded version of an artificial neural network, which is inspired by the human brain and effectively analyzes large amounts of data to discover patterns and characteristics in the data. Recently, the performance of deep learning models has been proven in imaging fields, and CNNs are widely used deep learning models [48–50]. The CNN multiplies by sliding the kernel at the location of the input data, summing the values, and summarizing them into a single value to determine the data's characteristics. This process serves to extract the elements contained in the data. According to the input data, the CNN model is divided into Convolutional 2D (Conv2D) and Convolutional 1D (Conv1D), and can be expressed as shown in Figure 4. Conv2D moves each pixel in the image data and derives a feature map using convolution, and Conv1D moves between variables and the batch size to derive a feature map using convolution. In this study, since time series data were used as input, we tried to derive rainfall characteristics using Conv1D. Conv1D is mainly used when dimension of input data is 1D data (such as sequential data), and Conv2D is used when dimension of input data is 2D data (such as image). In addition, when the input data is sequential data, the Long Shot Term Memory (LSTM) model can be also used. That is to say, the LSTM model is suitable when the input data is highly dependent on the past state. The input data of this study is sequential data, but the influence on external factors can be greater than the dependence on the past state. Therefore, in this study, it was judged that the CNN model, which can find the characteristics of data well from external factors, was suitable.

Teleconnection

Teleconnection is the theory that shows correlations between geographically distant regions; it states that one phenomenon can influence other phenomena due to spatial

distance and temporal difference. For example, the Pacific North America Pattern during El Niño strengthens the North Pacific High and Aleutian Lows, and it moves the Aleutian Lows eastward. During this process, the jet stream in the upper Pacific atmosphere also expands and strengthens eastward. Eventually, these pressures affect the mid-latitude atmospheric circulation.

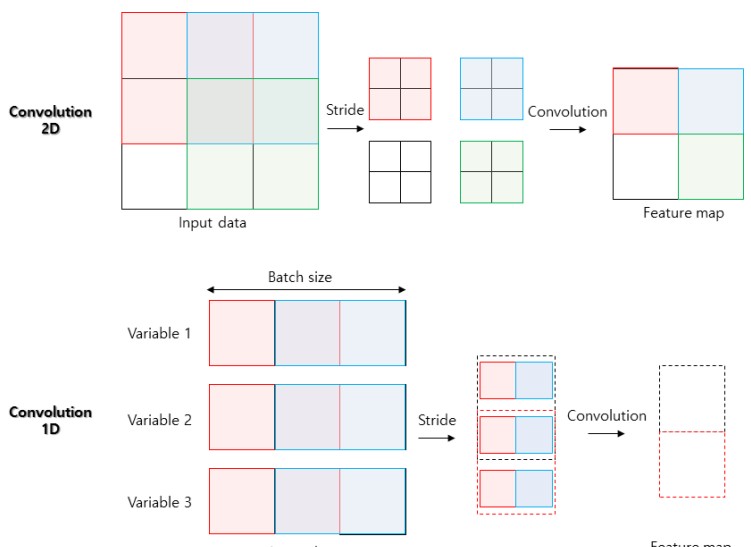

**Figure 4.** Concept of the CNN model and the difference between Conv1D and 2D.

GCMs are known to better reproduce global-scale atmospheric variability than detailed regional-scale climate distributions. Therefore, instead of using regional forecast data, GCMs improve the forecast by finding factors that either directly or indirectly affect the local climate predict the local climate. Teleconnection is a technique that can predict future rainfall or temperature in a target area by applying a lag-time between a regional-scale climate pattern and rainfall or temperature in the target area. Many studies have conducted teleconnection analysis to determine the spatial and temporal effects on the climate of the target area [51,52]. Summer precipitation in Korea, the target region in this study, is affected by the East Asian monsoon precipitation, which is related to El Niño-related Equatorial East Pacific Sea Surface Temperature [53,54]. In addition, the precipitation in this region is known to be highly correlated with sea surface temperature in India [55,56]. In this study, teleconnection analysis was performed using sea surface temperature as a meteorological factor that affects rainfall in Korea.

### 2.4. Evaluation Metrics

To evaluate the performance for each downscaling method in this study, this study used the mean squared error (MSE), root mean squared error (RMSE), and correlation coefficient (CC) as evaluation metrics, as shown in Equations (2)–(4).

$$\text{MSE} = \frac{1}{n}\sum_{i=1}^{n}(X_s - X_0)^2 \tag{2}$$

$$\text{RMSE} = \sqrt{\frac{1}{n}\sum_{i=1}^{n}(X_s - X_0)^2} \tag{3}$$

$$\text{CC} = \frac{\sum_{i=1}^{n}(X_0 - \overline{X}_0)(X_s - \overline{X}_s)}{\sqrt{\sum_{i=1}^{n}(X_o - \overline{X}_o)^2}\sqrt{\sum_{i=1}^{n}(X_s - \overline{X}_s)^2}} \tag{4}$$

where $X_s$ is the simulated data, $X_0$ is the observed data, $n$ is the total number of data, $\overline{X}_o$ is the average of the observed data, and $\overline{X}_s$ is the average of the simulated data.

## 3. Results

### 3.1. Application of Quantile Mapping (QM)

The reference data have different observation start dates (Incheon in 1904, Seoul in 1907, and Suwon in 1964). Thus, the models were trained from the observation start dates of each station to December 1989 based on quantile mapping. Figure 5 illustrates the results of QM training. The left side of Figure 5 is the cumulative probability distribution (CDF) of the raw GCM data and the reference data; the solid black line expresses the CDF for the reference data (obs. precipitation). The dashed blue line indicates the CDF for the GCM data. The right side of Figure 5 shows the CDF of the calibrated GCM data using the QM method as well as the CDF of the reference data. It can be observed that the CDF after calibration is similar to the CDF of the observed precipitation. The CDF before correction is different from the CDF of the reference data; however, it can be confirmed that the CDF after correction is almost the same. This result indicates that the statistics of the GCM data after calibration are similar to those of the reference data.

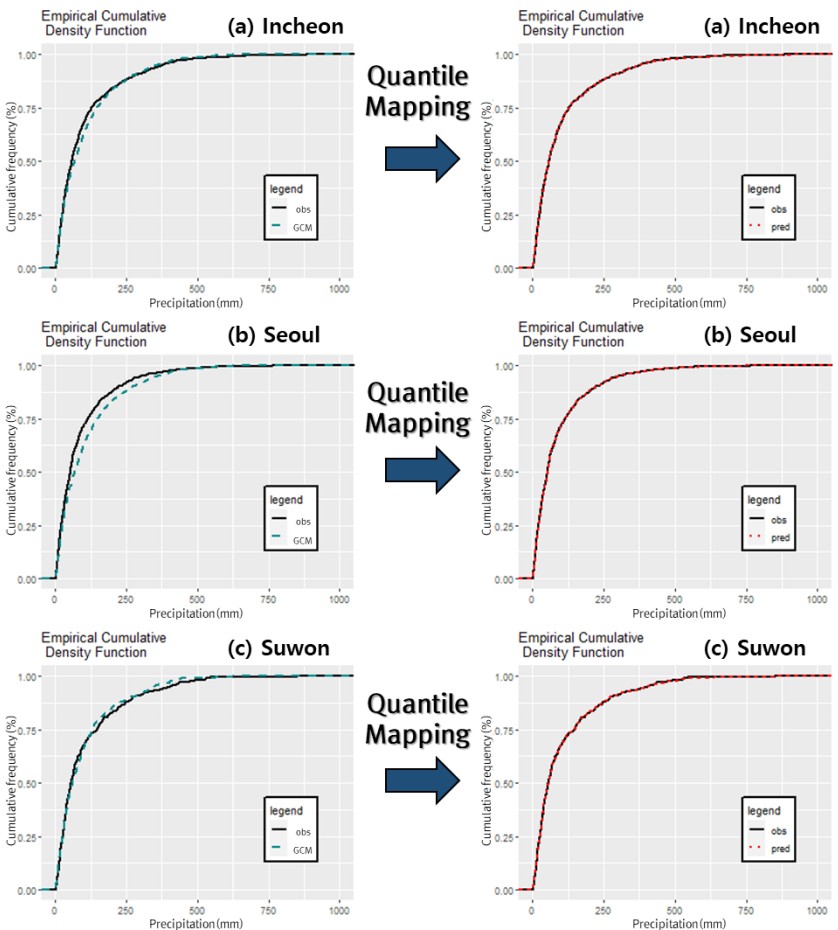

**Figure 5.** CDF before and after calibration of the GCM using QM and reference data (subfigure: (**a**) is Incheon, (**b**) is Seoul, (**c**) is Suwon).

Figure 6 shows the time series of the reference data and the calibrated GCM data using the QM method in the training set. Here, the blue lines are the time series of the reference data, and the dashed red lines are the time series of the calibrated GCM data. This figure shows that the range of values is similar to that of the reference data when compared with Figure 2; however, the calibrated GCM data show the same rainfall pattern for each observation station. In the figures for Incheon (a), Seoul (b), and Suwon (c), the solid and the dashed boxes show almost the same result in terms of the precipitation patterns, although the range of values differs. This result indicates that the QM method can calibrate

the range of values but that the patterns of precipitation cannot be calibrated. In addition, when the spatial downscaling of multiple stations corresponds to one grid, there is a limit to the spatial variability for each station.

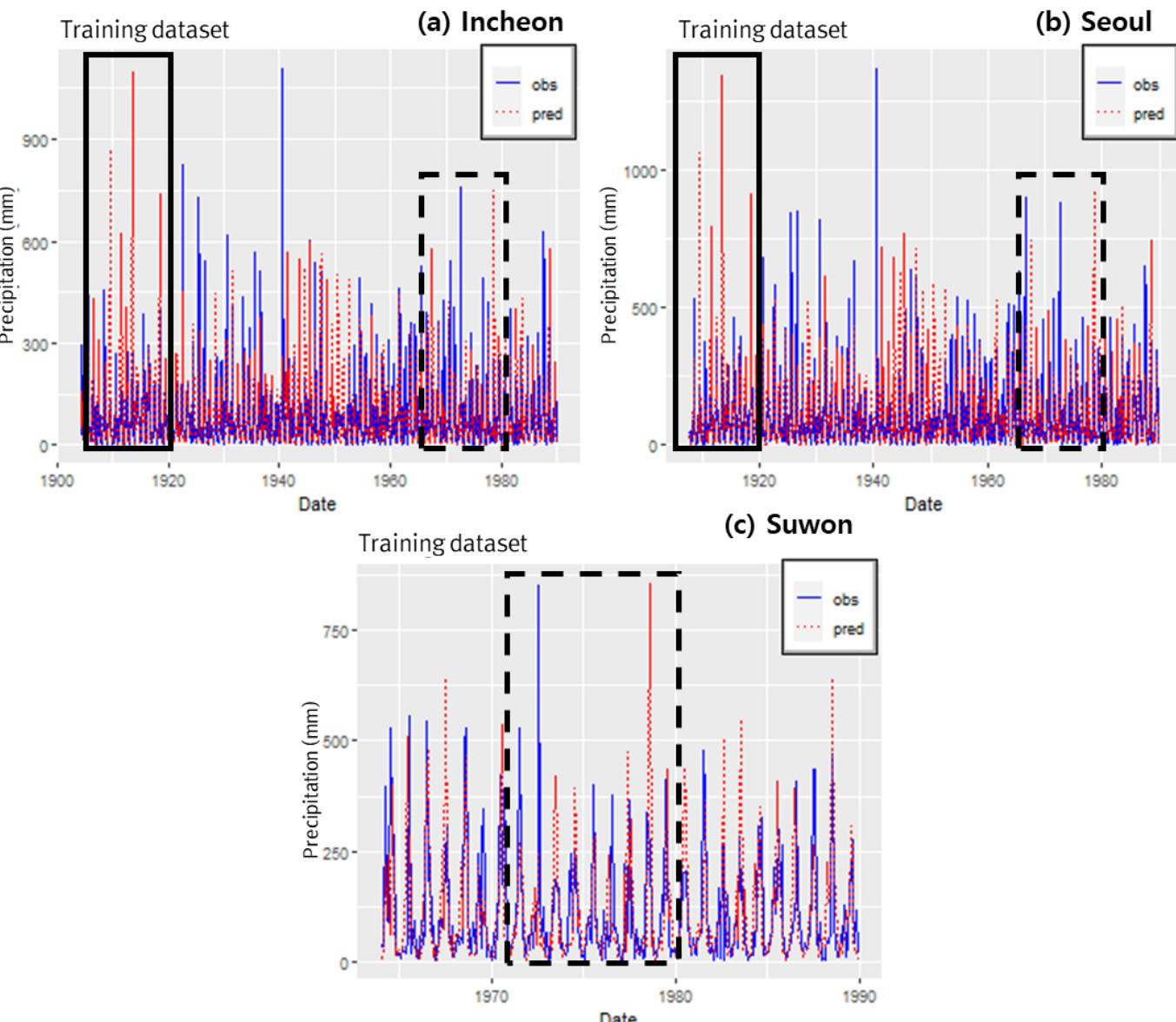

**Figure 6.** Monthly time series of the reference data and calibrated GCM data using QM (subfigure: (**a**) is Incheon, (**b**) is Seoul, (**c**) is Suwon).

### 3.2. Application of the CNN

The QM method can only correct the probability of occurrence using the cumulative probability distribution, while the CNN method has the advantage that it can consider other variables related to precipitation in the GCM data. In this study, teleconnection analysis was performed to extract the grid affecting the reference data in the global data, and the data of the selected grid was used as an explanatory variable.

#### 3.2.1. Teleconnection Analysis Using SST

In general, it is known that the temperature of the ocean affects the circulation of the atmosphere; this effect is related to extreme rainfall and drought events. Therefore, in this study, we tried to extract the grid highly correlated with each reference precipitation using SST data. Here, the temporal correlation was analyzed using a delay of up to approximately

12 months, and spatial correlation was analyzed across the global grid. The results of the analysis show that, when delayed by 1 month, the positive correlation was approximately 68% and the negative correlation was 60%. Thereafter, the correlation decreased until a delay of 5 months. Additionally, it was confirmed that the greatest correlation was found when the SST was delayed for 6 months. Therefore, for this study, the best correlation grid (positive and negative) was selected from among the SST data delayed by 6 months; this grid was used as an explanatory variable for CNN modeling. Figure 7 shows a 6-month delay among the teleconnection analysis results. Here, the red points represent the grids with the best positive correlation, and the blue points represent the grids with the best negative correlation. For Incheon, the best positive correlation was 71%, and the most negative correlation was 66%. For Seoul, the best positive correlation was 68%, and the most negative correlation was 64%. For Suwon, the best positive correlation was 74%, and the most negative correlation was 68%.

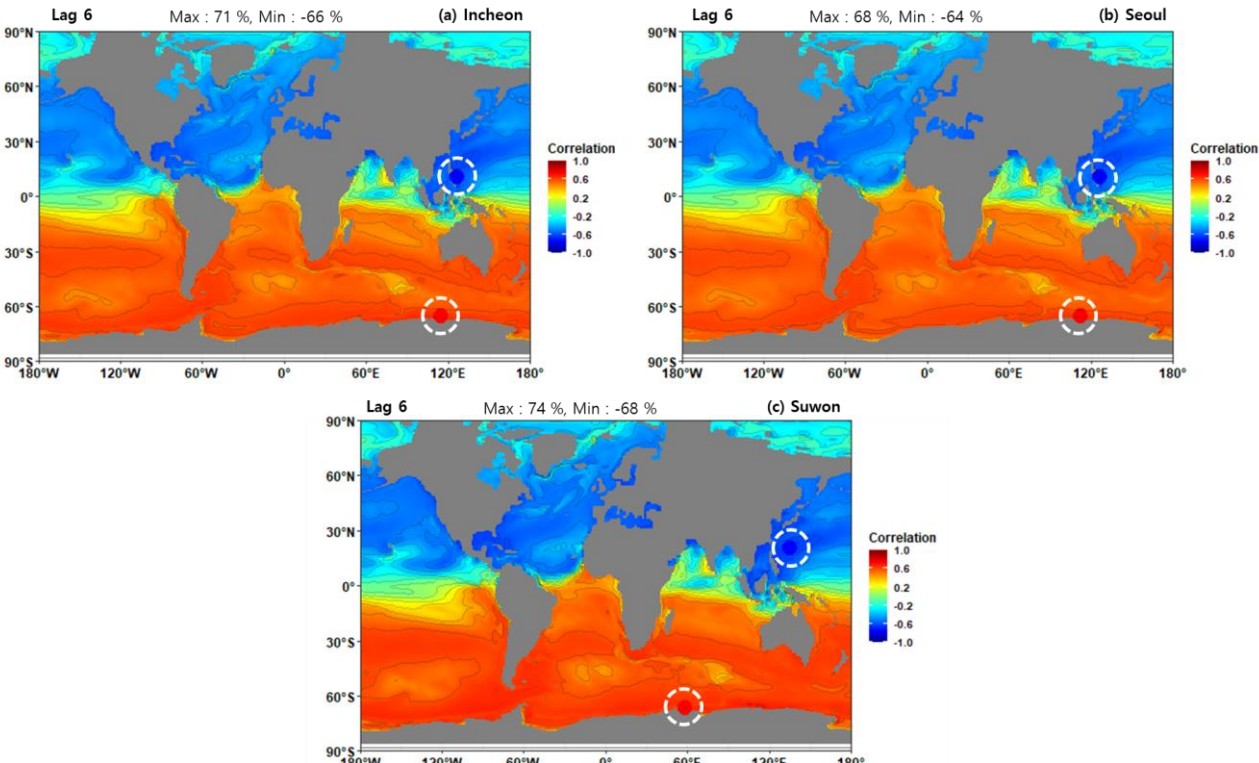

**Figure 7.** Result of teleconnection analysis for SST delayed by 6 months.

Table 4 summarizes the locations of the selected grids. The most positively correlated grids were located in the Indian Ocean, and the most negatively correlated grids were located in the Pacific Ocean. Many studies suggest that the SSTs in the Indian and Pacific Oceans influence the El Niño/Southern Oscillation [53–57]. This relationship also changes the monsoon variability, which affects Korea; therefore, the SST of the two selected regions may affect the rainfall in Korea.

### 3.2.2. CNN Model Training

We used the CNN method as a second method for spatial downscaling. To apply the CNN method, it is necessary to identify and collect explanatory variables. In this study, the SST data from the grid selected through teleconnection analysis and the meteorological data from the grid corresponding to the target area were defined as explanatory variables. Table 5 summarizes the explanatory and target variable used for CNN model training in this study.

**Table 4.** Locations of selected grids.

| Station | Most Positively Correlated Grid | Most Negatively Correlated Grid |
|---|---|---|
| Incheon | Longitude: 114° E<br>Latitude: 65° S | Longitude: 126° E<br>Latitude: 10° N |
| Seoul | Longitude: 112° E<br>Latitude: 65° S | Longitude: 126° E<br>Latitude: 10° N |
| Suwon | Longitude: 58° E<br>Latitude: 66° S | Longitude: 135° E<br>Latitude: 20° N |

**Table 5.** Description of target and explanatory variables used in this study.

| Variable | Abbreviation | Description |
|---|---|---|
| Target variable | Obs. pr | Reference precipitation data for each station |
| Explanatory variable | Clt | Cloud area fraction of the GCM |
| | Hurs | Relative humidity of the GCM |
| | Pr | Precipitation of the GCM |
| | Prw | Atmosphere mass content of water vapor of the GCM |
| | Ps | Surface air pressure |
| | Psl | Sea level pressure |
| | Tas | Air temperature of the GCM |
| | SST+ | Most positively correlated SST |
| | SST- | Most negatively correlated SST |

In this study, we performed an analysis to find correlations between the explanatory variables and target variables in Table 6. As a result, all explanatory variables had more than 50% correlation with the reference data. In addition, it was confirmed that the SST data had the greatest correlation. This result means that the global climate phenomenon has a greater influence on the reference data than the precipitation of the target grid.

**Table 6.** Result of the correlation between explanatory variables and target variable.

| Station | Clt | Hurs | Pr | Prw | Ps | Psl | Tas | SST+ | SST- |
|---|---|---|---|---|---|---|---|---|---|
| Incheon | 56% | 47% | 52% | 69% | −59% | −59% | 57% | 71% | −66% |
| Seoul | 55% | 47% | 51% | 68% | −58% | −58% | 57% | 68% | −66% |
| Suwon | 56% | 47% | 51% | 71% | −60% | −60% | 60% | 74% | −68% |

Figure 8 shows the architecture of the CNN model used in this study. This architecture consists of four Conv1D layers, four activation function layers, two dropout layers, a pooling layer, a flattening layer, and two dense layers. Here, the dropout layer was constructed to prevent overfitting, and the leaky relay layer was constructed to solve the phenomenon of dying neurons.

Table 7 summarizes the hyperparameters of the CNN model used in this study. Since the purpose of this study is a regression task, not a classification task, the loss function used was MSE. For the activation function, Relu was used in the convolution layer, and linear was used as the activation function in the output layer. The kernel sizes of the four convolutional layers have a constant value of 3. Adam was used as an optimizer function; it was chosen because it had the best effect when compared with optimizer functions in previous studies [58]. The batch size and the number of epochs were set to 128 and 100, respectively. When training CNN models in this study, 20% of the data from the batch size were used for validation. Figure 9 shows the optimization results of the CNN models. Here,

the red lines represent losses of training data and blue lines represent losses of validation data. As a result of training, the loss decreased as the number of epochs was repeated, and three CNN models were well-trained without overfitting. When the number of epochs for the three models was 50, the losses converged near the minimum. Figure 10 shows the time series of the reference data and the calibrated GCM data in the training set. Here, the blue lines represent the time series of the reference data, and the dashed red lines represent the time series of the calibrated GCM data.

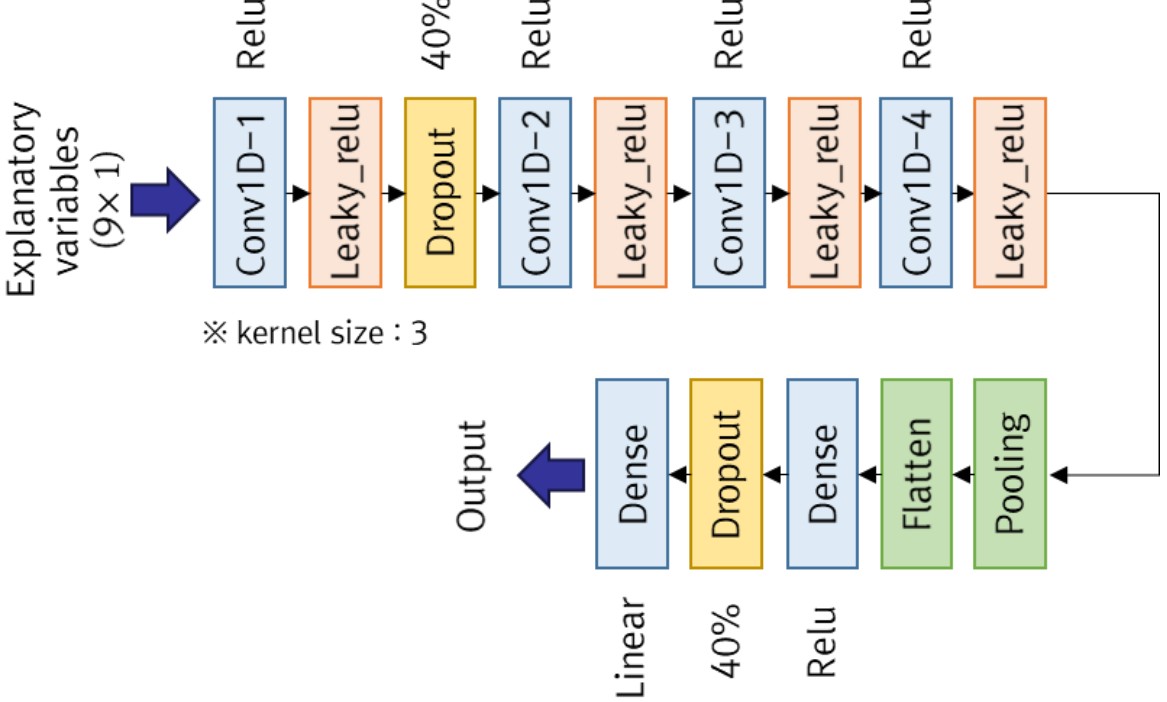

**Figure 8.** Architecture of the CNN model in this study.

**Table 7.** Hyperparameter of the CNN model used in this study.

| List | Parameter |
|---|---|
| Loss function | Mean Squared Error |
| Activation function | Conv1D layer: Relu, Output layer: Linear |
| Kernel size | 3 |
| Optimizer function | Adam |
| Batch size | 128 |
| Epoch | 100 |

Figure 10 shows that the range of values is similar to that of the reference data when compared with Figure 2. In addition, the precipitation pattern for each station is well-reflected. In particular, precipitation was well simulated in terms of extreme events such as the 1940 event at the Incheon station and the 1974 event at the Suwon station. A similar precipitation pattern to those shown in the boxes in Figure 5 is not found in Figure 10. Thus, when spatial downscaling of the GCM data is performed for precipitation, the CNN model proposed in this study showed better results for both the range of values and the precipitation pattern. In addition, this result shows that the CNN model is more effective than the traditional downscaling method, the QM method. However, since this result was produced by learning on the training set, it is difficult to see it as a systematic evaluation for each model. In systematic evaluation, it is necessary to apply and evaluate new data not used for model learning.

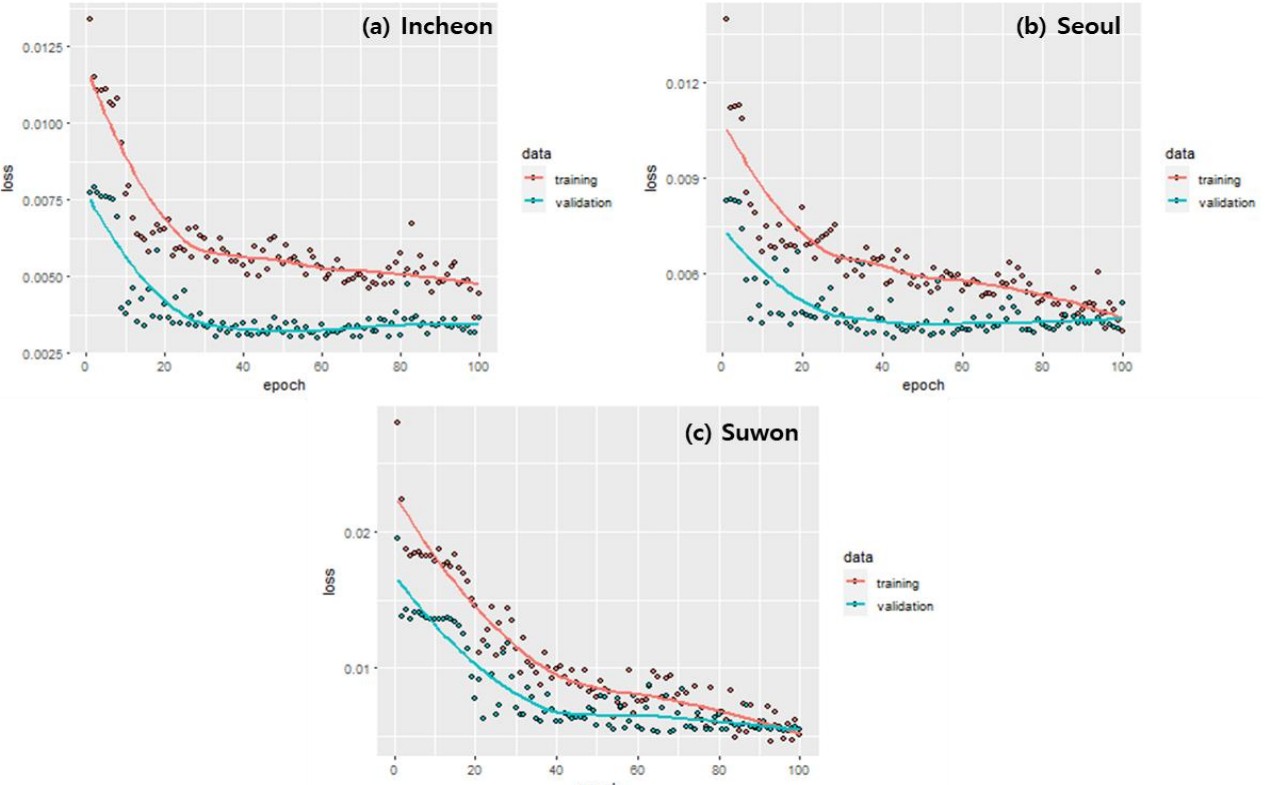

**Figure 9.** Result of CNN model optimization.

### 3.3. Performance Evaluation for Each Method

This section describes an accuracy evaluation that we performed using the two methods suggested in Sections 3.1 and 3.2, and the model showing the best performance was selected. The accuracy was evaluated using a test dataset (1990–2014) that was not used for model learning.

Figure 11 shows the time series for each method in the test dataset. Here, the solid red lines represent the reference data, the dashed blue lines represent the calibrated data using the CNN, and the dashed green lines represent the calibrated data using QM. In Figure 11, the precipitation estimated by the QM method was greater than the observed precipitation in 1998, which means that the QM method has limitations in predicting precipitation patterns as mentioned in Section 3.1. In addition, all the green lines represent overestimations compared to the reference data. On the other hand, the red lines, which represent the reference data, were accurate according to each station's precipitation pattern.

Table 8 summarizes the evaluation metrics used to compare each method. Here, the evaluation metrics included the CC and RMSE. The CNN model was found to be better than QM for all stations, showing better results in terms of both CC and RMSE. The CNN showed an average performance of approximately 69% CC and 117 mm RMSE. QM showed an average performance of approximately 41% CC and 169.9 mm RMSE. In terms of the CC, the performance improved by approximately 28% on average, and in terms of the RMSE, the error decreased by approximately 52 mm.

Taylor diagrams are mathematical diagrams designed to graphically indicate RMSE, CC, and Standard deviation (SD) [59]. Figure 12 shows Taylor diagram for each region. The green dotted line indicates RMSE, blue SD, black CC. The green square point indicates SD of observation. In general, the smaller the RMSE and SD, and the closer CC is to 1, which means the better model performance. The red and blue square points represent the performance of QM and CNN in the diagram. Here, it shows that the model of a point (red or blue square point) closer to the green square point in this diagram has better

performance. From this, we can see that the blue points are closer to the green square point than red points.

Thus, when spatial downscaling of the GCM data for precipitation was performed, the CNN model was more effective than the QM method (i.e., the traditional downscaling method). Therefore, the CNN model was selected as the final model to forecast future precipitation in this study.

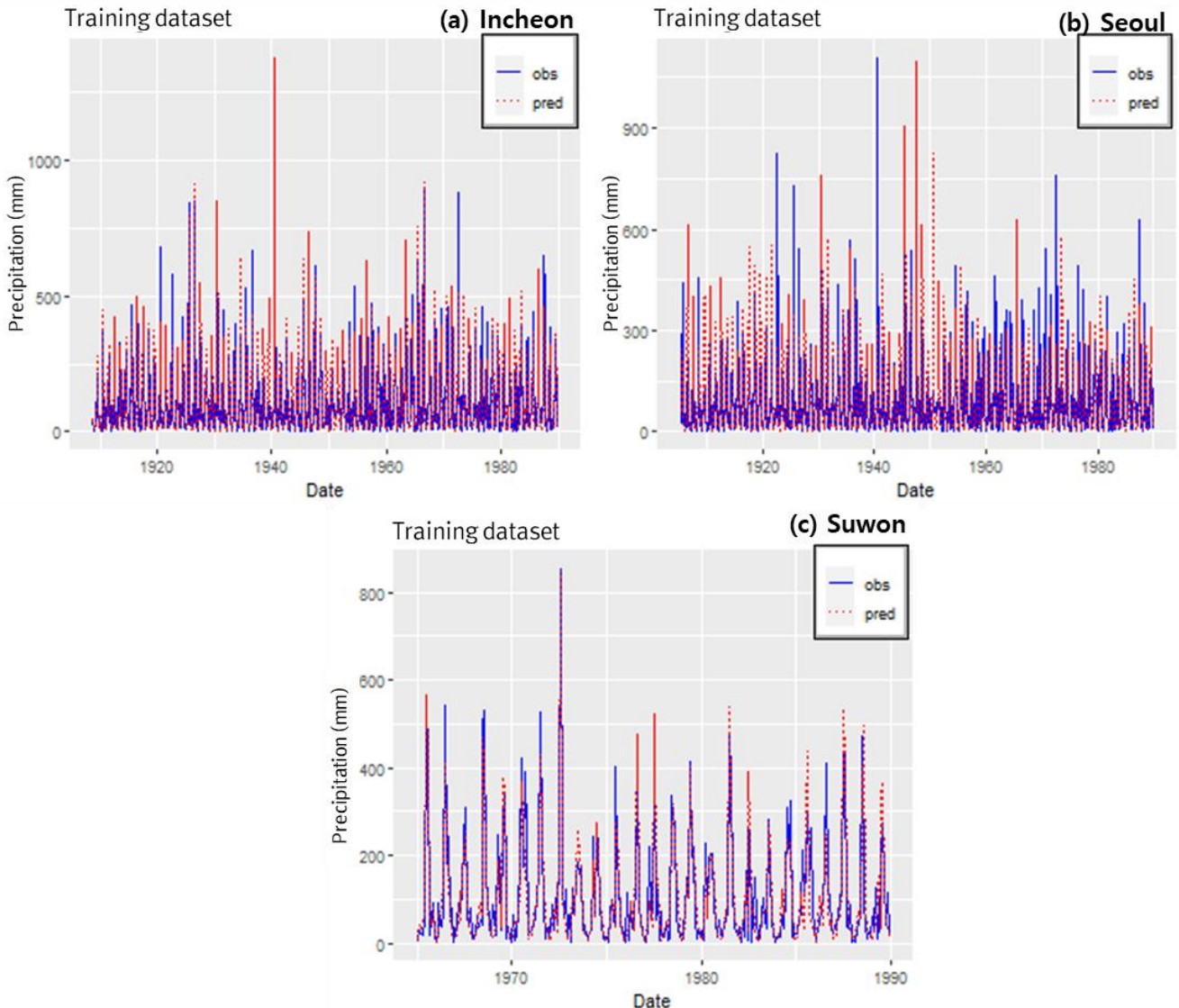

**Figure 10.** Monthly time series of the reference data and calibrated GCM data (using the CNN model).

**Table 8.** Evaluation metrics comparison for each method.

| Stations | QM | | CNN | |
|---|---|---|---|---|
| | CC (%) | RMSE | CC (%) | RMSE |
| Incheon | 36.68 | 199.67 mm | 60.92% | 146.92 mm |
| Seoul | 40.87 | 152.55 mm | 72.71% | 102.16 mm |
| Suwon | 45.25 | 157.64 mm | 73.35% | 102.46 mm |

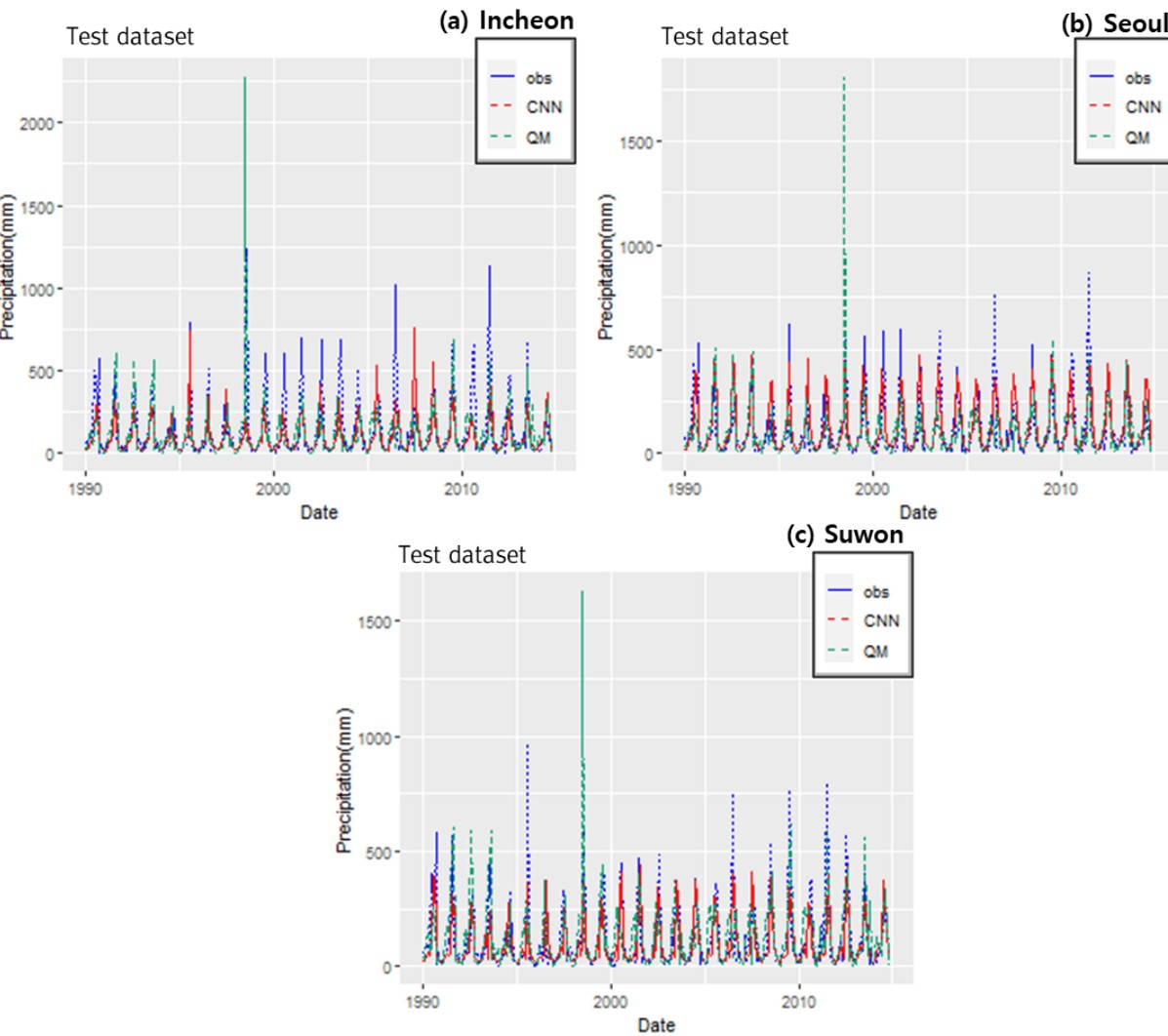

**Figure 11.** Time series of the calibrated GCM data by each method in the test set.

*3.4. Forecasting Precipitation Using the Final Model*

We used a climate change scenario and global data from GCMs to predict future weather patterns. This study downscaled the climate change scenario by reflecting the precipitation characteristics of three precipitation stations located in Korea. The downscaling methods are the QM method and the CNN method; the CNN method, which showed high accuracy on the training and test sets, was determined to be the optimal method. To determine how much future precipitation was affected by climate change at the three observatories, the future precipitation was forecasted in the short- (2015–2040), medium- (2041–2070), and long-term (2071–2100). Figure 13 and Table 9 show these results. As shown in Table 9, the precipitation at the Incheon station is expected to increase by approximately 3.3% by 2040, 14.3% by 2070, and 12.7% by 2100 compared to the present. Regarding the precipitation at the Seoul station, the predicted increase in precipitation was about 15–30% higher than the other two stations, and the precipitation at the Suwon station is predicted to decrease compared to the present. The 8.5 scenario used in this study is an extreme scenario in which greenhouse gas emissions maintain their current trend. In other words, if the current greenhouse gas emissions are maintained, rainfall may increase by 35% in 2100. Therefore, it is necessary to establish a disaster prevention plan to reduce natural disasters in the future and develop mitigation measures to reduce greenhouse gas emissions.

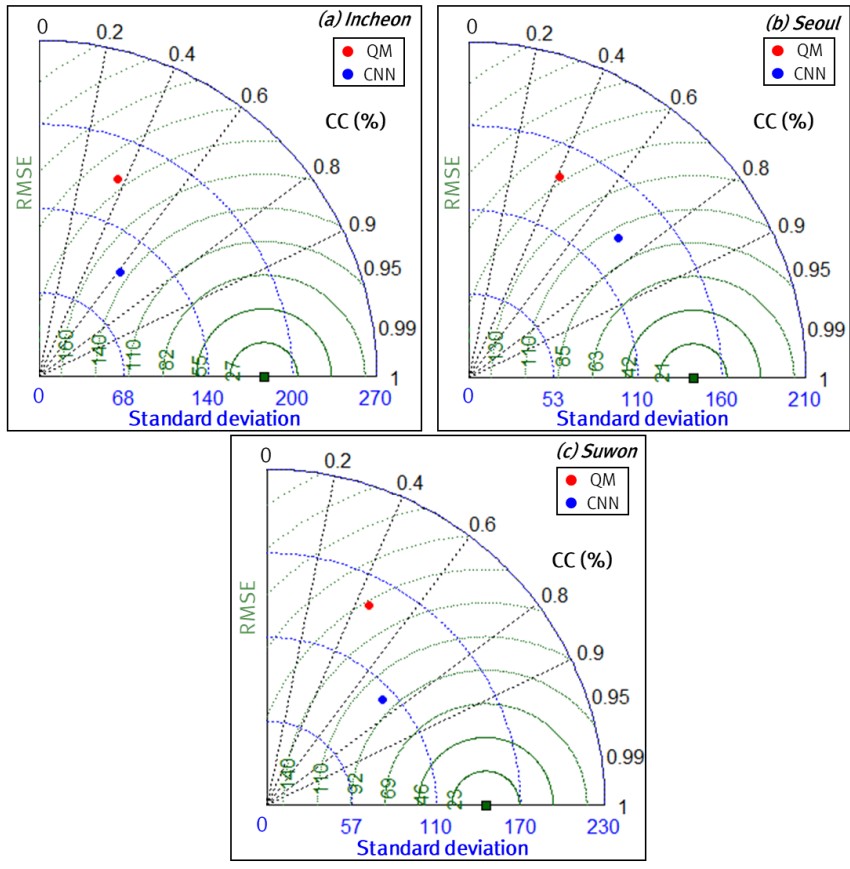

**Figure 12.** Taylor diagram for each region: (**a**) Incheon, (**b**) Seoul, and (**c**) Suwon.

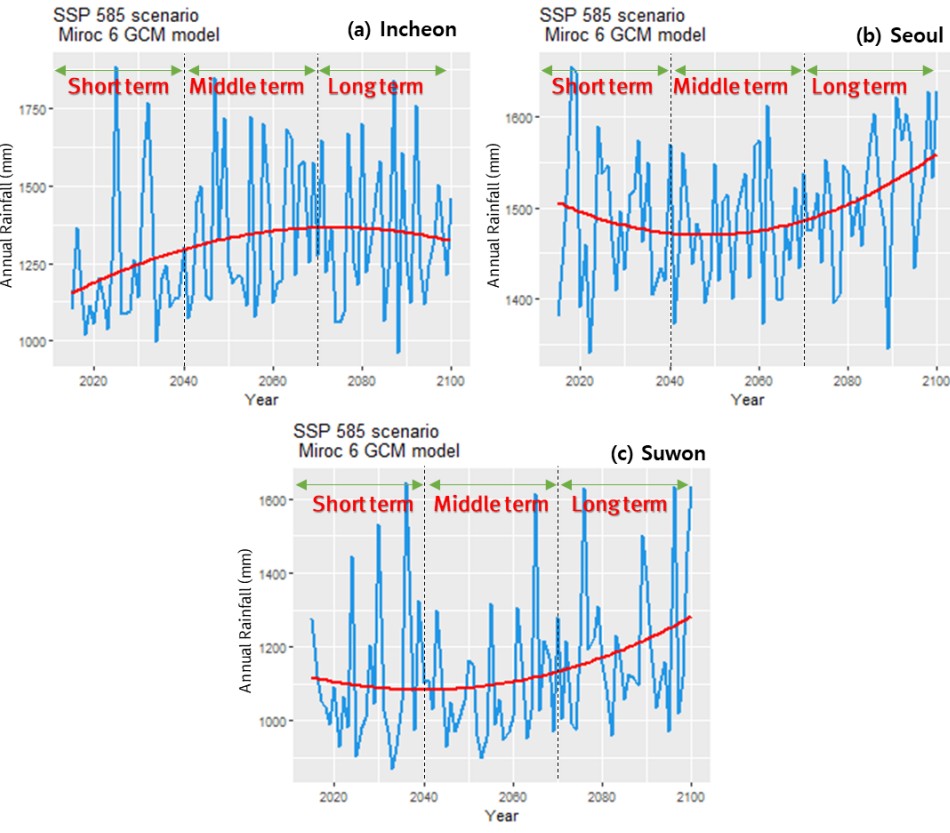

**Figure 13.** Future precipitation forecasting time series using the final model.

**Table 9.** Rate of increase and decrease for future average precipitation forecasting.

| Station | Reference Data (For 30 Years) | 2015–2040 (Short-Term) | 2041–2070 (Medium-Term) | 2071–2100 (Long-Term) |
|---|---|---|---|---|
| Incheon | 1187.5 mm | 1227.0 mm | 1356.8 mm | 1338.4 mm |
| | | 3.3% | 14.3% | 12.7% |
| Seoul | 1125.5 mm | 1486.4 mm | 1475.9 mm | 1516.6 mm |
| | | 32.1% | 31.1% | 34.7% |
| Suwon | 1177.9 mm | 1109.7 mm | 1091.63 mm | 1198.2 mm |
| | | −5.8% | −7.3% | 1.7% |

## 4. Conclusions and Discussions

This study proposes a CNN model considering teleconnection for the spatial downscaling of precipitation in GCM data. The CNN model can effectively find complex characteristics using many different variables, and teleconnection can find global climate phenomena that affect the precipitation in a target area. Therefore, in this study, a highly correlated spot was derived from the global grid through teleconnection and the gridded meteorological data were used as the independent variables. Then, the CNN model was developed for spatial downscaling considering the independent variables. The proposed method was compared with QM, which has been traditionally used as a statistical downscaling method.

When applying the QM method, the precipitation range of the GCM data was effectively corrected. However, the precipitation patterns at three stations (Incheon, Seoul, and Suwon) were the same, which means that the QM method has a limitation in that it cannot consider the spatial variability within one grid. Meanwhile, the CNN method calibrated the range and value of the precipitation better than QM. To compare these results, the models were evaluated on test set data that were not used for model learning. The CNN showed an average performance of approximately 69% CC and 117 mm RMSE, while QM showed an average performance of approximately 41% CC and 169.9 mm RMSE. Thus, the CNN model was more effective than the QM method. Finally, as a result of forecasting the precipitation until 2100 using the CNN method, the average precipitation increased by 16.7% compared to the reference data.

The CNN model considering teleconnection was more effective than the QM method at correcting the precipitation pattern and range of the GCM data. However, since it focused on spatial downscaling, there is a limitation that temporal downscaling is not suitable. Thus, we will study temporal downscaling methods in the future. In addition, this study only applied MIROC6 from among the various GCM data. Therefore, it is necessary to confirm the applicability of the method proposed in this study by using various GCM data such as that from NOAA, GISS, and CCSM in the future. Additionally, since this study focuses on developing a downscaling method that can consider spatial variability even in a single grid, a relatively narrow area was set as the target area. Therefore, the target area should be expanded to extensively evaluate the applicability of the method proposed in future studies. Therefore, we write Case Study in the title of the paper.

This study suggested an improved spatial downscaling technique and confirmed that the proposed method could effectively supplement the bias in GCM data. Therefore, the results of this study can be used for hydrological analysis considering climate change; furthermore, it is expected that it can be utilized in disaster prevention plans and urban planning.

**Author Contributions:** Conceptualization, J.K., M.L., and H.H.; data curation, J.K. and D.K.; formal analysis, J.K.; funding acquisition, H.S.K.; methodology, J.K. and M.L.; supervision, H.S.K.; visualization, Y.B.; writing—original draft, J.K. and M.L.; writing—review and editing, H.H. All authors have read and agreed to the published version of the manuscript.

**Funding:** INHA UNIVERSITY Research Grant 66357.

**Institutional Review Board Statement:** Not applicable.

**Informed Consent Statement:** Not applicable.

**Data Availability Statement:** Not applicable.

**Acknowledgments:** This work was supported by INHA UNIVERSITY Research Grant.

**Conflicts of Interest:** The authors declare no conflict of interest.

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
