# Peer review of "Case Study: Development of the CNN Model Considering Teleconnection for Spatial Downscaling of Precipitation in a Climate Change Scenario"

_sustainability, doi:10.3390/su14084719_

Round 1

Reviewer 1 Report

Review of sustainability-1658215: " Development of the CNN model considering teleconnection for spatial downscaling of precipitation in a climate change scenario".

I have received the manuscript sustainability-1658215 for reviewing. A new spatial downscaling method is proposed in this paper. The results are significant. However, my main concern is the applicability of this method, as this study only analyzed three sites in a very small area. Therefore, I think this manuscript should do a major revision before acceptance for publication. I would like to give my specific comments as following:

  1. Line 13: GCM data are often used to analyze future climate change rather than “how climate change will affect the future climate.”
  2. Abbreviations are usually defined when they first appear in the main text and then are used in the remainder of the manuscript.
  3. In line 197, the author did not indicate the types of meteorological data. How did the authors choose the type of data? Or take all the data together as the input data for CNN model. In other words. In another word, please specify the dimension of the input data of the convolutional neural network.
  4. In Line 198, the starting time of the training set is not written.
  5. Fig. 6 and 11. I suggest the author use a Taylor diagram to represent the result.
  6. Fig. 7. Why did the author include global results instead of regional (study area) results? What is the significance of minimum (negatively) value for this study?
  7. In Fig 8, Why is the dropout layer only set after the first convolutional layer? What is the reason for this setting? For example, can we set dropout layers after the second and third convolutional layers?
  8. The size of the convolution kernel in Fig. 8 is not stated in the text. Are the kernel sizes of the four convolutional layers consistent?
  9. The author applied 1D-CNN model for time series analysis. Theoretically, RNN based models such as LSTM are more suitable for time series related simulation. Please explain the advantages of CNN model compared with other deep learning methods, especially RNN models such as LSTM.
  10. Lines 365-366. This study only analyzes the applicability of CNN in a very small area, and the three stations are in the same grid. Although the differences among the three stations are compared, the applicability of the method remains to be discussed.

Author Response

Thank you very much for your insightful and detail review. We have revised attentively the manuscript in order to include your comments. We believe that this manuscript is substantially improved through the result of the revision. Please see our point-by-point response (in blue) to your comments (in red). In addition, response for your main concern is written on comment 10. The sites are limited and so we write Case Study in the title of the paper

Reviewer 2 Report

First at all, congratulations for the interesting work. Only one petition, improve the explanation of the CNN procedure.

Some format mistakes:

  • Line 13. "Global climate models (GCM)..." --> Be carefully with details (GCMs).
  • Line 36. Avoid value judgment like "serious natural disasters"
  • Line 45. The sentence could be improved, maybe "...published in 2007 (AR4) and 2014 (AR5)..."
  • Line 45.  "and a new AR6 assessment report..."
  • Line 46.  "by these assessment report" --> Change for "by these AR"
  • Line 119.  "peninsula at 33-38º" Error (38º)? Check
  • Figure 1. Error on the Grid (Y axis), start in 35ºN (the 35ºN is repeated).
  • Line 133. "MIROC6climate" --> Change for "MIROC6 climate"
  • Table 1. The time period conclude in 12-01 and not in 12-31? 
  • Table 2. "Location" column --> Include units.
  • Table 3. The data are referenced to monthly step? Specify.
  • Table 3. Use thousands separator (homogeneity).
  • Figure 2. (Same comment to Figure 10).
    • Increase the size of the independent figures. Are difficult to analyse.
    • Y axis. "Precipitation(mm)" Include space between varaible and it unit. "Precipitation (mm)".
    • "gcm" legend --> Use capital letter "GCM"
    • a) Incheon -->  "Date" axis --> 2021?
    • Figure title. "Monthly time series..."
  • Line 175. "quantile mapping" --> Change for "QM"
  • Line 198. Use the same format for the dates --> Change "." for "-"
  • Line 206 and 222. Use consecutive numbering. Change:
    • "1. Convolutional Neural Network" --> "2.3.2.1 Convolutional Neural Network"
    • "2. Teleconnection" --> "2.3.2.2 Teleconnection"
  • Line  210. "Convolutional Neural Network (CNN)"
  • Improve the definition of:
    • Conv1D
    • Conv2D
  • Figure 5. 
    • Y axis "Cumulative frequency (%)"
    • Legend "gcm" --> Capital letter--> "GCM"
    • X axis "Precipitation(mm)" --> "Precipitation (mm)"
  • Paragraph (lines 317 - 324). Partially repeated.
  • Line 324. "target variables"
  • Table 6. Title "...target variables". 
  • Figure 8, Table 7 and others. Reference before it appears.
  • Improve the definition of:
    • Relu function.
    • Adam function.
  • Table 8. Column "CC" of QM. Include the unit.
  • Figure 12. Axis Y "Annual Rainfall" --> Include the unit.
  • Table 9. Define the variables included in the table (mean?)
  • Line 428. "quantile mapping" Change for "QM".

Author Response

Thank you for your favorable review of this manuscript. We have revised attentively the manuscript in order to include your comments. In addition, we improved the explanation of the CNN procedure in manuscript to reflect your comments.

Reviewer 3 Report

This article have new innovation in methods and results.

Author Response

Thank you for your favorable review of this manuscript.

Reviewer 4 Report

The  paper  is well  written, but, the author may do some revisions as

  • Authors are recommended to cite all references in text carefully. For example, in page 2, line 60 “(Xu et al. [17]…) .” should be  “(Yang et al. [17]…).” , also, in page 2, line 71 “ Xu et al. [19]”   should be   “( Xu & Wang [19]”. Please check all of them, not only my example.
  • In page 2, line 91, the word “Recently,…”does not match the two [36,37] which belong to 2011, 2012, respectively.
  • In page 8, line 246 “in Equation (2)–(4).” Should be “in Equations (2)–(4).” .
  • Some of the sentences are incomprehensible and the research needs proofreading.
  • Put dot (.) at the end of the sentences in the citation part of the all figures.
  • Conclusions and Discussions not in short and crispy.

Author Response

Thank you for your favorable review of this manuscript. We have revised attentively the manuscript in order to include your comments. Please see our point-by-point response (in blue) to your comments (in red).

Round 2

Reviewer 1 Report

I think this manuscript is suitable for publication.